# Spontaneous exciton dissociation enables spin state interconversion in delayed fluorescence organic semiconductors

Alexander J. Gillett [1✉], Claire Tonnelé [2], Giacomo Londi [3], Gaetano Ricci[4], Manon Catherin[5], Darcy M. L. Unson[1], David Casanova [2], Frédéric Castet [6], Yoann Olivier[4], Weimin M. Chen [7], Elena Zaborova[5], Emrys W. Evans [1,8], Bluebell H. Drummond [1], Patrick J. Conaghan[1,9], Lin-Song Cui[1,10], Neil C. Greenham [1], Yuttapoom Puttisong [7✉], Frédéric Fages [5✉], David Beljonne [3✉] & Richard H. Friend [1✉]

Engineering a low singlet-triplet energy gap ($\Delta E_{ST}$) is necessary for efficient reverse inter-system crossing (rISC) in delayed fluorescence (DF) organic semiconductors but results in a small radiative rate that limits performance in LEDs. Here, we study a model DF material, BF2, that exhibits a strong optical absorption (absorption coefficient $= 3.8 \times 10^5 \, cm^{-1}$) and a relatively large $\Delta E_{ST}$ of 0.2 eV. In isolated BF2 molecules, intramolecular rISC is slow (delayed lifetime $= 260 \, \mu s$), but in aggregated films, BF2 generates intermolecular charge transfer (inter-CT) states on picosecond timescales. In contrast to the microsecond intramolecular rISC that is promoted by spin-orbit interactions in most isolated DF molecules, photoluminescence-detected magnetic resonance shows that these inter-CT states undergo rISC mediated by hyperfine interactions on a ~24 ns timescale and have an average electron-hole separation of ≥1.5 nm. Transfer back to the emissive singlet exciton then enables efficient DF and LED operation. Thus, access to these inter-CT states, which is possible even at low BF2 doping concentrations of 4 wt%, resolves the conflicting requirements of fast radiative emission and low $\Delta E_{ST}$ in organic DF emitters.

[1] Cavendish Laboratory, University of Cambridge, JJ Thomson Avenue, Cambridge, UK. [2] Donostia International Physics Centre (DIPC), Donostia, Euskadi, Spain. [3] Laboratory for Chemistry of Novel Materials, Université de Mons, Place du Parc 20, 7000 Mons, Belgium. [4] Unité de Chimie Physique Théorique et Structurale & Laboratoire de Physique du Solide, Namur Institute of Structured Matter, Université de Namur, B-5000 Namur, Belgium. [5] Aix Marseille Univ, CNRS, CINaM UMR 7325, AMUtech, Campus de Luminy, 13288 Marseille, France. [6] Institut des Sciences Moléculaires, Université de Bordeaux, 33405 Talence, France. [7] Department of Physics, Chemistry and Biology (IFM) Linköping University, Linköping, Sweden. [8] Department of Chemistry, Swansea University, Singleton Park, Swansea, UK. [9] ARC Centre of Excellence in Exciton Science, School of Chemistry, University of Sydney, Sydney, NSW 2006, Australia. [10] Department of Polymer Science and Engineering, University of Science and Technology of China, Hefei, Anhui 230026, China. ✉email: ajg216@cam.ac.uk; yuttapoom.puttisong@liu.se; frederic.fages@univ-amu.fr; david.beljonne@umons.ac.be; rhf10@cam.ac.uk

In organic light-emitting diodes (OLEDs), the spin statistical recombination of injected electrons and holes leads to the formation of a 3:1 ratio of spin-triplet to spin-singlet excitons[1]. Since triplet excitons in organic emitters are optically dark, the internal quantum efficiency (IQE) of an OLED is restricted to 25% if no further steps are taken to utilise triplet excitons for light emission. One approach is to develop organic semiconductors that can use a reverse intersystem crossing (rISC) process to convert these dark triplet states into bright singlet excitons, assisted by ambient thermal energy[2]. This class of materials, known as thermally activated delayed fluorescence (DF) emitters, enable OLEDs, with IQEs approaching 100%[3]. To obtain a singlet-triplet energy gap ($\Delta E_{ST}$) small enough for an efficient rISC process (typically <0.1 eV), it is necessary to weaken the electron exchange interaction. To achieve this, DF emitters generally comprise of spatially separated electron donor (D) and acceptor (A) moieties, resulting in excitons with intramolecular charge transfer (intra-CT) character[4]. Consequently, rISC can proceed through spin-orbit interactions, often also involving triplet states localised on either the D or A ($^3$LE) that are in close energetic proximity to the intra-CT excitations[5–8]. However, DF emitters with intra-CT-type excitons suffer from a low oscillator strength and can only operate efficiently if competing non-radiative processes are highly suppressed[9]. As a result, it is challenging to create DF materials that achieve the necessary balance between the conflicting requirements of a small $\Delta E_{ST}$ for effective rISC and a large oscillator strength for rapid and efficient emission. Therefore, it is desirable to design new classes of organic semiconductors that can combine the strong absorption and emission of localised excitons with the weak exchange interaction of electronic excitations with spatially separated electrons and holes.

We show here that it is possible for the intra-CT excitations of DF emitters to spontaneously dissociate on picosecond timescales, even in highly dilute films, into loosely bound intermolecular charge transfer (inter-CT) states. Due to the greatly reduced $\Delta E_{ST}$ in these well-separated electron-hole pairs, the hyperfine interaction (HFI) can mediate the spin state interconversion between the singlet and triplet inter-CT manifolds. Subsequently, reformation of the bright intramolecular singlet exciton results in efficient DF. We demonstrate that this is the dominant spin state interconversion mechanism in a model DF emitter, BF2, which possesses an extremely high oscillator strength for its lowest-energy intramolecular singlet exciton. Thus, we establish a mechanism of OLED operation that combines the desirable optical attributes of conventional fluorescent emitters and the spin manipulation abilities of DF organic semiconductors.

## Results

We study four DF emitters (Fig. 1), selected to provide a range of exchange energies and intra-CT transition oscillator strengths,

from large (BF2) to small (TXO-TPA), see Supplementary Figs. 1 and 2 and Supplementary Table 1. We begin by focussing on BF2, which supports efficient OLED operation in the near infrared region[10]. BF2 comprises of two triphenylamine D units and a boron difluoride-based A core, forming a DAD structural motif. Consequently, the excitons of BF2 possess a partial intra-CT character, though the electron-hole overlap is still sufficient to retain a high absorption coefficient of $3.8 \times 10^5$ cm$^{-1}$ (Supplementary Fig. 1) and a moderate $\Delta E_{ST}$ of 0.2 eV (Supplementary Fig. 3)[10]. Furthermore, the large ground-state dipole moment of BF2 curcuminoid derivatives drives the formation of closely interacting dimers[10–12], even in dilute films, opening up the possibility for strong intermolecular interactions in the solid state.

**Optical spectroscopy studies of BF2**. To investigate the excited-state dynamics of BF2, we have performed ultrafast transient absorption (TA) spectroscopy on a neat BF2 film (Fig. 2a–b). In the TA spectra at 0.2–0.3 ps, we observe two positive bands centred at 630 and 740 nm and a negative photo-induced absorption (PIA) feature peaking at 1000 nm. We assign the positive bands at 630 and 740 nm to the ground-state bleach (GSB) and stimulated emission (SE) of BF2, respectively (see Supplementary Fig. 4 for steady-state absorption and photoluminescence). The PIA at 1000 nm is present immediately after excitation and is due to the spin-singlet intra-CT exciton (intra-$^1$CT) of BF2. The SE and intra-$^1$CT PIA decay on picosecond timescales, demonstrating that the bright intra-$^1$CT excitons are being lost, but there is very little decrease in the GSB region on these same timescales. In the kinetic taken from the high-energy GSB edge (570–630 nm) to avoid overlap with the decaying SE, the GSB intensity remains unchanged at 10 ps, whilst the SE has decreased to 30% of peak intensity. Thus, the process quenching intra-$^1$CT excitons on these timescales is efficient but does not result in decay back to the ground-state. As the forward ISC rates for organic DF emitters are typically ~$10^8$–$10^7$ s$^{-1}$, we can rule out ISC to the triplet manifold as the reason for the loss of the intra-$^1$CT on picosecond timescales[6,13–15]. We notice the formation of a new PIA band at 950 nm; the growth of this species mirrors the loss of the SE, signifying it is being formed from intra-$^1$CT excitons. The new species with a PIA at 950 nm is the hole polaron on BF2, as seen in the TA of a BF2:PC$_{60}$BM 1:1 blend film (Supplementary Fig. 5); here, the efficient photocurrent generation in a reference organic solar-cell (OSC) device fabricated from this system confirms that electron transfer is induced from BF2 to PC$_{60}$BM (Supplementary Fig. 6), leaving behind a hole on BF2. We therefore show that intermolecular charge transfer takes place in neat BF2 on picosecond timescales. This is consistent with our observations in OSC devices fabricated with a neat BF2 active layer (Supplementary Fig. 7a) where we obtain a moderate short-circuit current density of 0.43 mA/cm$^2$ under AM1.5 G 1 sun illumination (Supplementary Table 2), and a peak photovoltaic external quantum efficiency (EQE$_{PV}$) of 2.1% at 555 nm under short-circuit conditions (Supplementary Fig. 7c). We find that the EQE$_{PV}$ obtained is limited by the relatively short excited-state lifetime in neat BF2 (Supplementary Fig. 11h), where only a small fraction of the initially photogenerated excitations remain by 1 µs, a timescale that is typical for charge extraction in OSCs[16].

We also find that there can be efficient regeneration of the emissive intra-$^1$CT excitons in BF2 from the spatially separated charge carriers at longer times. This is most easily observed when BF2 is diluted in the non-interacting OLED host material CBP (4,4′-Bis(N-carbazolyl)-1,1′-biphenyl), which allows for control over the intermolecular interactions and kinetics[10]. In Fig. 2c–d, we present the nanosecond-microsecond TA of BF2 diluted at 10

**Fig. 1 Chemical structures.** The chemical structures of the four model delayed fluorescence materials investigated in this study.

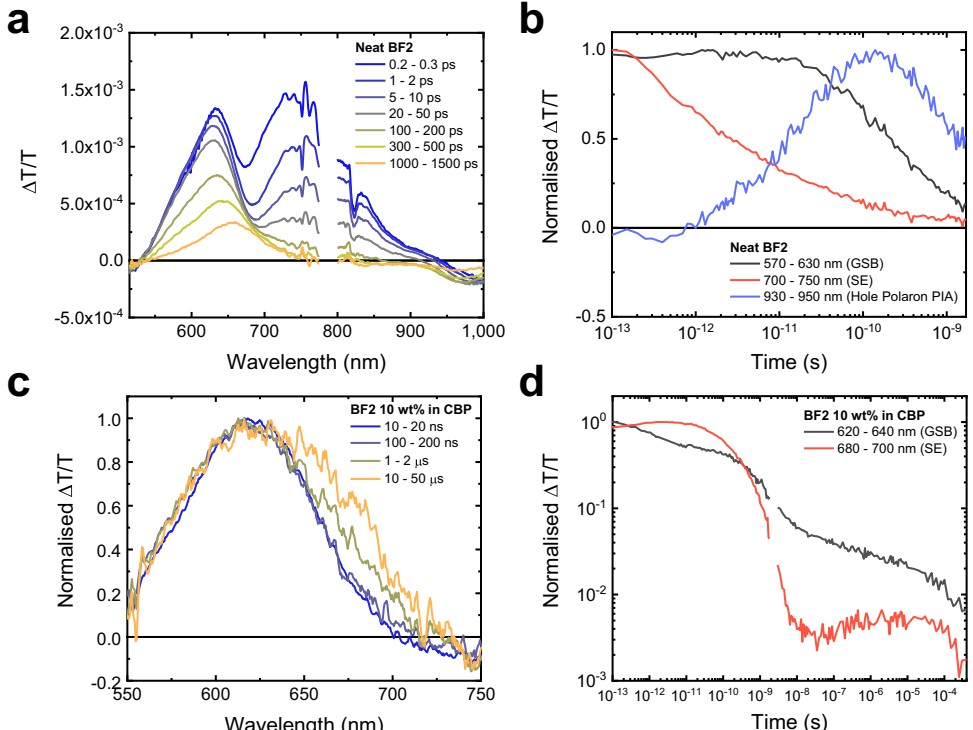

**Fig. 2 Transient absorption studies of BF2 films. a** The ultrafast transient absorption spectra of a neat BF2 film, excited at 610 nm with a fluence of 7.0 μJ cm$^{-2}$. The disconnect in the spectral data is due to the gap in our probe range around the 800 nm fundamental of the laser used to run the setup. **b** The transient absorption kinetics of the neat BF2 film, taken from the ground-state bleach (GSB; 570–630 nm), stimulated emission (SE; 700–750 nm) and hole polaron (930–950 nm) regions. (**c**) The normalised nanosecond transient absorption spectra of BF2 doped at 10 wt% in CBP, excited at 532 nm with a fluence of 15.7 μJ cm$^{-2}$. (**d**) The combined kinetics of the ultrafast and nanosecond-microsecond transient absorption of BF2 doped at 10 wt% in CBP, taken from the GSB (620–640 nm) and SE (680–700 nm) regions. 532 nm excitation was used for both measurements, with a fluence of 15.7 μJ cm$^{-2}$ for the ultrafast and 346 μJ cm$^{-2}$ for the nanosecond transient absorption. The kinetics were scaled relative to each other by their respective fluences. The disconnect in the kinetics is due to the gap between the time ranges probed by the ultrafast and nanosecond TA measurements.

weight percentage (wt%) in CBP. Here, we observe the delayed regrowth of the SE band between 650–725 nm over microsecond timescales, matching closely the blue-shifted SE of dilute BF2 in the corresponding ultrafast TA measurements (Supplementary Fig. 13). In addition, we show that the emissive species present on the timescales of SE reformation has the same photoluminescence (PL) spectrum as the intra-$^1$CT of BF2 (Supplementary Fig. 15b), confirming that these states are being regenerated. These findings indicate that the electronic excited states of BF2 can readily interconvert between the localised intramolecular excitations, loosely bound singlet inter-CT (inter-$^1$CT) states and even free charge carriers. Similar behaviour is observed in the polar host DPEPO (Bis[2-(diphenylphosphino)phenyl]ether oxide), which has a higher dielectric constant of ε = 6.1[17] compared to ε = 3.5 for CBP[18], implying that the surrounding dielectric environment does not have a significant effect on the dissociation of intra-$^1$CT excitons in BF2 (Supplementary Figs. 16–18). As discussed later, the efficient interconversion between the intra-$^1$CT and inter-$^1$CT can be rationalised by electronic structure calculations that suggest strong electronic coupling and a small energy gap between them (Supplementary Table 10).

We consider that this spontaneous exciton dissociation plays a critical role in OLEDs employing BF2 in the emissive layer. The thermally assisted intramolecular rISC process that is used to convert dark triplet excitons into bright singlet states in organic DF emitters is typically promoted by spin-orbit interactions, often involving intermediate triplet states[5–8]. In-line with a previous report[10], our calculations reveal the presence of the second lowest triplet exciton (T$_2$) in between the lowest-energy intra-$^1$CT and

intra-$^3$CT excitations in the BF2 monomer (Supplementary Table 6), leading to an endergonic rISC that can rationalise the very long (260 μs) delayed lifetime of isolated BF2 molecules (Supplementary Fig. 11b). However, the formation of inter-CT states with spatially separated electrons and holes in aggregated BF2 will greatly reduce the strength of the electron exchange interaction, bringing in near-resonance the singlet and triplet states, as shown by electronic structure calculations (Supplementary Fig. 37). Thus, it becomes possible for the HFI to mediate the interconversion of loosely bound inter-CT states with singlet and triplet character in aggregated BF2. Typical HFI-ISC rates are on the order of ~10$^8$–10$^6$ s$^{-1}$, enabling efficient spin state interconversion via a periodic oscillation between the $M_S = 0$ singlet ($^1$CT$_0$) and the $M_S = -1, 0, +1$ triplet sublevels ($^3$CT$_-$, $^3$CT$_0$, $^3$CT$_+$) of an inter-CT state (Fig. 3a), otherwise known as a spin-correlated radical pair[19–22].

**Magnetic resonance studies of BF2.** We utilise PL-detected magnetic resonance (PLDMR), performed at 293 K to ensure relevance to OLED device operation, to explore whether the HFI mediates the ISC processes in BF2. In the PLDMR of a neat BF2 film (Fig. 3d, full field range data in Supplementary Fig. 19), we observe a narrow negative signal at ~333 mT, which is attributed to inter-CT states[23]. To understand the relevance of this feature, we note that under an external magnetic field (**B$_0$** ~333 mT), the Zeeman interaction decouples HFI-induced transitions between the inter-$^1$CT$_0$ and either inter-$^3$CT$_+$ or inter-$^3$CT$_-$[24]. Consequently, microwave spin-pumping from the inter-$^3$CT$_0$ (formed

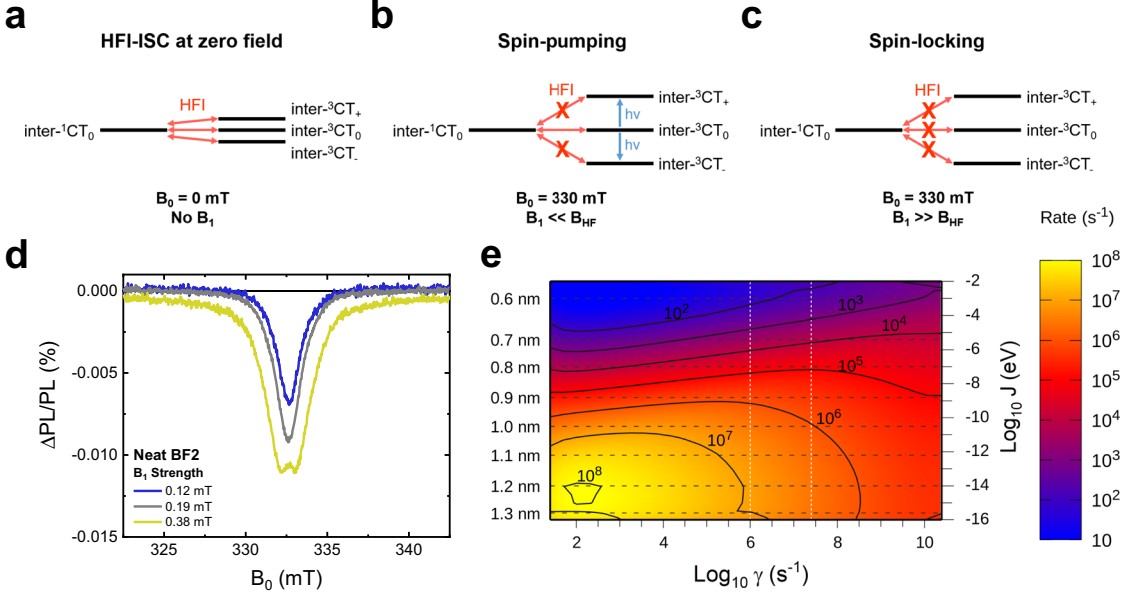

**Fig. 3 Magnetic resonance studies of hyperfine couplings in a BF2 film. a** A schematic demonstrating the ability of the HFI to mediate spin-mixing processes between the $M_S = 0$ inter-$^1$CT and the $M_S = -1,0,+1$ inter-$^3$CT sublevels in the absence of an external magnetic field. **b** Under an applied external magnetic field $\mathbf{B_O}$, the Zeeman interaction energetically forbids HFI-induced transitions between the inter-$^3$CT$_+$ and inter-$^3$CT$_-$ and the inter-$^1$CT$_0$. Spin-pumping $M_S = \pm 1$ transitions can then occur between the inter-$^3$CT$_0$ and the inter-$^3$CT$_+$ and inter-$^3$CT$_-$ sublevels; this reduces the inter-$^3$CT population that can ultimately couple to the emissive intra-$^1$CT manifold via the inter-$^1$CT state, resulting in a decreased PL yield from the sample and the sharp, negative PLDMR signal seen at ~333 mT. **c** When the magnetic field of the applied microwaves $\mathbf{B_1}$ is perpendicular to $\mathbf{B_O}$ and becomes larger than the local hyperfine field $\mathbf{B_{HF}}$, spin-locking occurs. This reduces the rate of the HFI-mediated inter-$^1$CT$_0$ to inter-$^3$CT$_0$, transitions, locking the inter-CT state spin-population in the initially generated inter-$^1$CT$_0$. Spin-locking manifests as the formation of a characteristic W-shaped peak in the PLDMR as $\mathbf{B_1}$ is increased. **d** The PLDMR response of a neat BF2 film at 293 K with 405 nm excitation (30 mW). **e** The calculated HFI-ISC rate in a model BF2 dimer as a function of the dephasing rate, $\gamma$, and the electron exchange energy, $J$. The dashed white lines represent the experimental dephasing times in BF2 (upper bound = 40 ns, lower bound = 1 μs), as estimated from the TA measurements of BF2 at 10 wt% in CBP film (see SI for details). The left axis shows the electron-hole separation in the inter-CT state corresponding to each value of $J$.

via HFI-ISC from inter-$^1$CT$_0$) to inter-$^3$CT$_+$ and inter-$^3$CT$_-$ reduces the number of inter-$^3$CT$_0$ states that can couple to the singlet manifold, lowering the PL yield from the intra-$^1$CT (Fig. 3b). Therefore, the appearance of this feature provides strong evidence that: (i) HFI-ISC processes are occurring in loosely bound inter-CT states with a very small exchange energy (on the order of neV)[25]; and (ii) the nominally-dark inter-CT states can readily interconvert with the bright intra-$^1$CT excitons. From the full width at half maximum (FWHM) of the negative signal (1.5 mT), when the magnetic field of the applied microwave radiation ($\mathbf{B_1}$) is 0.12 mT, we estimate an upper bound of 0.75 mT (21 MHz) for the zero-field splitting $\mathbf{D}$-parameter, which is related to the average inter-spin distance[26]. $\mathbf{D} \leq 0.75$ mT corresponds to a lower bound of $\geq 1.5$ nm for the inter-CT state electron-hole radius ($r_{e-h}$) in neat BF2 (full calculation details in the SI)[27,28], far larger than expected for an intramolecular exciton[26]. We also observe the presence of this same negative signal in BF2 diluted at 10 wt% in CBP (Supplementary Fig. 20), confirming that HFI-ISC can also take place in the dilute films that are typically employed in OLED devices. Interestingly, the inter-CT PLDMR response is stronger in the 10 wt% blend than the neat BF2 film (Supplementary Fig. 19), which we attribute to the greater number of excited states living long enough to undergo HFI-ISC processes in the 10 wt% film (Supplementary Fig. 11).

Additional information about the HFI-ISC processes occurring in inter-CT states can be obtained by investigating the response of the PLDMR signal to increased microwave intensities, where $\mathbf{B_1}$ is perpendicular to $\mathbf{B_O}$. When $\mathbf{B_1}$ exceeds the hyperfine field ($\mathbf{B_{HF}}$, typically ~1 mT in organic radicals[20,29–31]), the rate of the HFI-mediated inter-$^1$CT$_0$/inter-$^3$CT$_0$ transitions is reduced as the radical spins begin to precess around $\mathbf{B_1}$ with the same frequency, negating their HFI-induced spin precession frequency difference[25,32,33]. In the absence of an effective spin-mixing process, the spin population now becomes trapped in the initially generated inter-$^1$CT$_0$, leading to spin-locking (Fig. 3c). Spin-locking results in a reduction of the number of inter-$^3$CT$_0$ states formed, so the spin-pumping in the triplet manifold becomes less efficient. Consequently, the PL yield begins to rise again under a large $\mathbf{B_1}$, manifesting as the formation of a broader resonance with a characteristic W-shaped peak in the PLDMR[25]. We observe the formation of this W-shaped signal at a $\mathbf{B_1}$ of 0.38 mT in the neat BF2 film. The effective $\mathbf{B_{HF}}$ can be estimated from twice $\mathbf{B_1}$ at the onset of spin-locking, and we obtain an effective $\mathbf{B_{HF}}$ of ~0.76 mT; this corresponds to a HFI-induced periodic singlet-triplet interconversion time of ~24 ns in the inter-CT states of BF2 (full calculation details in the SI). Thus, due to the large $\Delta E_{ST}$ and slow rISC process of isolated BF2 molecules, we propose that HFI-ISC between inter-CT states is the dominant triplet to singlet interconversion process in OLED devices fabricated from BF2. Furthermore, the singlet-triplet interconversion time of ~24 ns and $r_{e-h}$ of $\geq 1.5$ nm in BF2 suggest that the HFI-ISC processes observed may be in the coherent regime[34].

**Effect of dopant concentration on the intermolecular mechanism.** To better understand the interplay between the intra- and intermolecular rISC mechanisms BF2, we have performed further studies on a BF2 dilution series in CBP. We first note that the photoluminescence quantum efficiency (PLQE), measured in a nitrogen environment, rises from 44.3 to 63.3% as

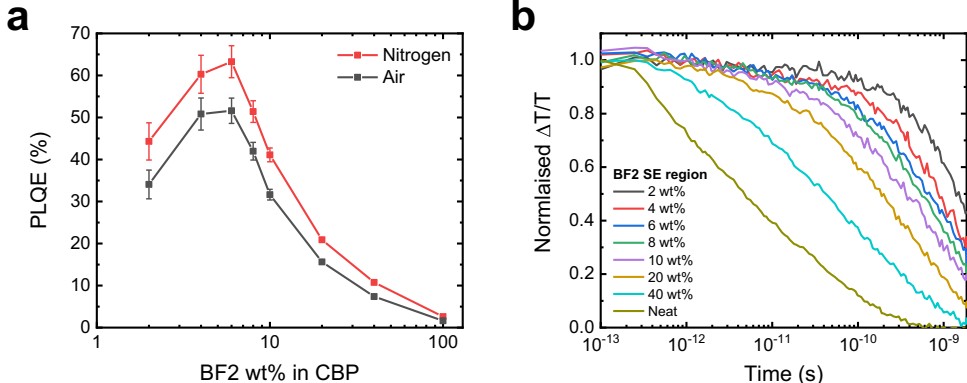

**Fig. 4 Effect of doping wt% on the photophysical behaviour of BF2 films. a** The photoluminescence quantum efficiency (PLQE) of BF2 doped in CBP at varying wt% (excitation 520 nm). The films were measured with and without exposure to the oxygen in ambient air, which results in the partial quenching of the delayed fluorescence contribution to the PLQE. The error bars represent the uncertainty in the PLQE measurements, as determined from the measured laser power fluctuations of 1% and the film absorption. **b** The normalised ultrafast transient absorption kinetics of the BF2 stimulated emission (SE), excited at 600 nm. Kinetic traces have been averaged between 680–730 nm for 2, 4, 6, 8, and 10 wt% films, 700–750 nm for 20 and 40 wt% films, and 720–770 nm for the neat BF2 film. Excitation fluences are provided in Supplementary Fig. 27.

the doping fraction is increased from 2 to 6 wt%, before decreasing again (Fig. 4a, tabulated PLQE values in Supplementary Table 3). We also find that the PLQE is consistently lower in films exposed to air than in those measured in a nitrogen environment, indicating a reduction in the DF contribution through the quenching of spin-triplet excitations by oxygen. Though the complete quenching of DF by oxygen in thin films is not expected due to incomplete oxygen penetration into the matrix[35], we make the empirical observation that the PL fraction quenched upon oxygen exposure rises as the BF2 concentration increases (Supplementary Fig. 26), implying a larger DF contribution to the observed PLQE in higher wt% films. The trend in PLQE with doping fraction is consistent with the reported electroluminescence external quantum efficiencies ($EQE_{EL}$) of BF2 OLEDs, where the best performance was found for BF2 at 6 wt% in CBP[10]. Such behaviour is unusual for DF emitters, as the PLQE and $EQE_{EL}$ are generally expected to fall as the doping fraction increases due to concentration quenching effects[36]. Furthermore, the PL maxima of the BF2 films shows a strong red shift with increasing concentration (Supplementary Fig. 8), meaning an additional reduction in the PLQE (and $EQE_{EL}$) is expected at higher wt% because of the energy gap law in organic emitters[37]. This effect is expected to be particularly severe when the peak emission wavelength shifts from 700 nm towards 800 nm[38], as occurs at higher BF2 loadings.

When examining the TA of the dilution series (Fig. 4b, corresponding TA spectra in Supplementary Fig. 27), we find a clear reduction in the SE lifetime with increasing concentration, which we have attributed to the formation of inter-$^1$CT states from the bright intra-$^1$CT exciton. Indeed, the noticeable decrease in the SE lifetime when moving from 2 to 4 wt% films suggests that there can be sufficient intermolecular interactions between emitter molecules to allow for the formation of inter-$^1$CT states at very low doping fractions; we note that the formation of inter-$^1$CT states at 4 wt% is also associated with a large increase in the film PLQE. Thus, we propose that in the 2 wt% films, the clear bi-exponential decay of the BF2 GSB, with a long delayed lifetime of 260 μs (Supplementary Fig. 11b), represents the intramolecular ISC/rISC mechanism of isolated BF2 molecules driven by spin-orbit interactions. Here, the relatively low PLQE is primarily ascribed to nonradiative losses occurring in the triplet manifold due to the slow and inefficient rISC process. When moving from 2 wt% to 6 wt%, the enhanced intra-$^1$CT quenching and increased

PLQE suggests the activation of the intermolecular HFI-ISC mechanism, where the rapid interconversion between inter-$^1$CT and inter-$^3$CT states reduces the nonradiative losses in the triplet manifold. Finally, further increasing the doping concentration results in a reduction of the PLQE again, likely due to the combined effects of longer-range charge separation over larger BF2 aggregates, which is associated with nonradiative recombination losses[39], as well as the enhanced nonradiative decay rates accompanying the strongly red shifted emission.

**Studies of other delayed fluorescence emitters**. We have explored three further model DF materials with smaller exchange energies and packing motifs that may or may not allow for the formation of inter-CT states: APDC-DTPA, TXO-TPA, and 4CzIPN[2,40,41]. In the first two materials, we observe a strong negative signal at ~333 mT in both the neat and dilute (10 wt%) films, with spin-locking also present in the neat films (Supplementary Figs. 21–25). This finding suggests that the dissociation of excitons is present in well-studied DF emitters in the solid state, with HFI-ISC between inter-CT states contributing to the triplet to singlet conversion in these OLED devices. We believe that this can be attributed to the close intermolecular interactions facilitated by the tendency of many DF materials to aggregate, even when diluted in a non-interacting host at a low wt%[10,42,43]. Additionally, many DF materials, including APDC-DTPA[40], also demonstrate excellent performance in non-doped OLEDs[40,44–51], with a growing number of DF emitters also exhibiting aggregation-induced DF[49–55]. In aggregation-induced DF, the DF properties are only present in environments with significant intermolecular contact, such as neat films; therefore, we propose that HFI-ISC between the inter-CT states, which requires close intermolecular contacts between the emitters, may be a significant rISC mechanism in these systems. In contrast, we observe a very weak inter-CT response in a neat 4CzIPN film and no spin-locking (Supplementary Fig. 25), demonstrating that intermolecular HFI-ISC is not a major singlet-triplet mixing process in this material. This is consistent with the low $EQE_{PV}$ (0.53%) of an OSC device fabricated with a neat 4CzIPN active layer (Supplementary Fig. 7c), which indicates that the excitons of 4CzIPN do not dissociate as readily as those in BF2. This is confirmed by the TA of a neat 4CzIPN film (Supplementary Fig. 28), where no evidence for intermolecular charge transfer is observed for timescales up to 2 ns, also supported by modelling (vide infra). As

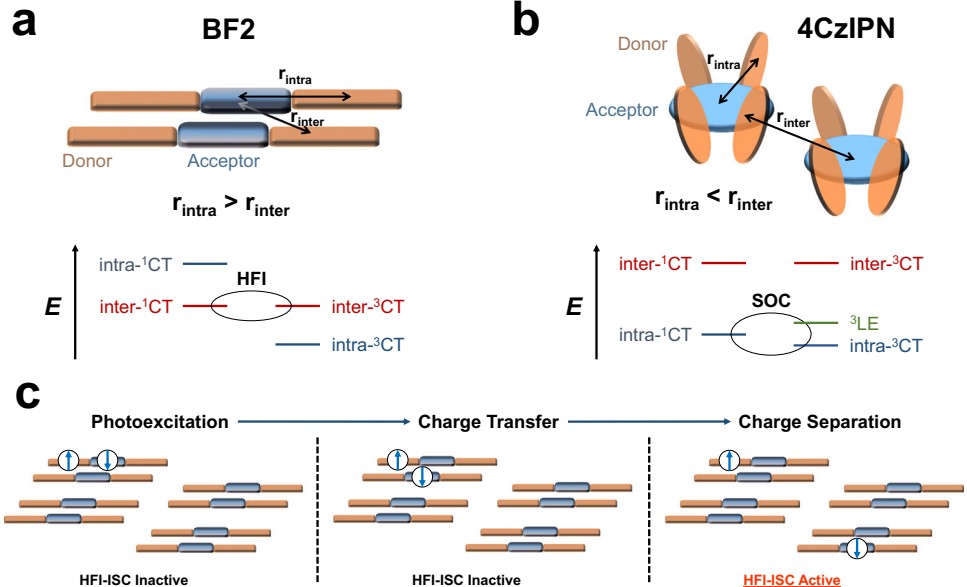

**Fig. 5 The role of intermolecular excitations in delayed fluorescence. a** A schematic of a representative BF2 dimer and the most relevant electronic excitations (state energies not to scale), demonstrating how the electron-hole separation in the inter-CT state can be less than the intra-CT exciton. This renders the inter-CT states more stable than the intra-$^1$CT, meaning intermolecular charge transfer following photoexcitation is thermodynamically favourable. The loosely bound inter-CT states can then undergo HFI-ISC processes, enabling efficient spin mixing in BF2, followed by recombination to the intra-$^1$CT for light emission. **b** A schematic of a representative 4CzIPN dimer and the most relevant electronic excitations (state energies not to scale), where the electron-hole separation in the inter-CT state is significantly larger than the intra-CT excitons. Consequently, the inter-$^1$CT state is >0.12 eV higher than the intra-$^1$CT excitons and intra-$^1$CT to inter-$^1$CT interconversion does not readily occur. As a result, the ISC processes in 4CzIPN are promoted by spin-orbit coupling, SOC, likely involving the intra-$^1$CT, intra-$^3$CT and $^3$LE states. **c** The photophysical processes occurring in a neat BF2 film that enable efficient spin state interconversion via the HFI. Following photoexcitation, the intra-$^1$CT dissociates to form an inter-$^1$CT state with the neighbouring molecule in the dimer. Subsequently, the inter-CT state can readily obtain longer-range separation, likely enabled by the shallow dependence of the inter-CT state energy on distance, to form loosely bound inter-CT states where the exchange energy is small enough for the HFI to mediate the ISC processes.

a result, the HFI-ISC channel we observe in the other DF emitters is largely unavailable to 4CzIPN and the ISC processes must proceed via spin-orbit interactions, as generally considered in the literature[6,27].

**Computational studies**. We have carried out computational studies of the lowest-energy electronic excitations of the representative cases of BF2 and 4CzIPN, which exhibit a contrasting ability to form inter-CT states. Our theoretical approach combines screened tuned range-separated hybrid functional time-dependent density functional theory (TDDFT) calculations with a (Boys) diabatization scheme that decomposes the electronic eigenstates of the system into a set of pure (diabatic) intra- and inter-CT electronic configurations, see SI for details. We begin by examining BF2 molecules embedded in a dielectric continuum. In line with earlier reports[10], our calculations show that the lowest intramolecular singlet excitation involves in-phase mixing of the $D^+(AD)^-$ and $(DA)^-D^+$ zwitterionic configurations and carries a very large oscillator strength (Supplementary Table 3). The lowest-energy intramolecular triplet excitation (intra-$^3$CT) has a greater weighting on the central A unit than the intra-$^1$CT, leading to a relatively large $\Delta E_{ST} = 0.28$ eV (Supplementary Table 6), in good agreement with our experimental observations. To explore the inter-CT states in BF2, we first note that X-ray diffraction measurements on single crystals of the BF2 parent molecule show the presence of strongly interacting dimers (with an intermolecular separation of ~0.35 nm)[12], driven by electrostatic interactions between the large, short-axis polarised, ground-state molecular dipoles. Because these dimers likely experience different environments in the disordered films, we have performed calculations for a range of intermolecular distances ($d_\perp$)

from the crystal value of 0.35 to 0.7 nm (Supplementary Figs. 25–34).

In Fig. 5a, we summarise the nature of the lowest-energy intra- and intermolecular electronic excitations in BF2, and a full schematic of all the excitations present and their dependence on $d_\perp$ is included in the SI. The most striking result is the appearance of inter-$^1$CT states at a lower energy than the intra-$^1$CT exciton. This energy ordering of the electronic excitations in BF2 can be rationalised through the analysis of the excited-state wavefunctions, namely their corresponding electron-hole radii. For a broad range of intermolecular distances between 0.35–0.5 nm (Supplementary Fig. 35), the averaged electron-hole distance in the intra-$^1$CT exciton is larger than in the inter-$^1$CT states. This result supports the view that intermolecular charge transfer is thermodynamically favourable in closely interacting BF2 molecular dimers, in line with our experimental observations. Furthermore, we find that the dependence of the inter-$^1$CT state energy with $d_\perp$ is surprisingly weak; a simple fit with a $1/(\varepsilon \, d_\perp)$ Coulomb law of the inter-$^1$CT state energy would lead an unphysical value of $\varepsilon = 7.5$. This is because the effective electron-hole separation in the inter-$^1$CT state exceeds the intermolecular distance $d_\perp$ owing to a large intramolecular component (at $d_\perp = 0.35$ nm, $r_{e-h}$ is already ~0.6 nm). Thus, the intra-CT character of the exciton effectively crops the short-range part of the Coulomb potential for the inter-$^1$CT states. We expect this to favour the formation of loosely bound inter-CT states, or even free charge carriers, provided continuous pathways that enable long range charge separation are present. In contrast to the relatively large $\Delta E_{ST}$ values calculated for the intra-CT excitons, the $\Delta E_{ST}$ of the inter-CT states is vanishingly small due to the minimal electron-hole overlap. The energy ordering predicted in the triplet manifold is

now swapped compared to the singlet, with the lowest-energy triplet adiabatic excitation the intra-$^3$CT (Supplementary Table 14). This is because the intra-$^3$CT, with an increased contribution from the central A moiety, has an average $r_{e-h}$ of only 0.4 nm and is consequently more strongly bound than the inter-$^3$CT states.

The calculated inter-$^1$CT states in the crystal dimer have a $r_{e-h}$ of ~0.6 nm, which is less than half the value (~1.5 nm) estimated from the FWHM analysis of the PLDMR signal. Though Equation S1 provides only an approximate value for $r_{e-h}$, we conclude that spin conversion occurs either in dimers with intermolecular distances exceeding the equilibrium value in the crystal, or through non-nearest-neighbour interactions in larger aggregates. To investigate the latter hypothesis, we have conducted similar TDDFT calculations on a BF2 tetramer. The results reported in Supplementary Fig. 37 show that a broad manifold of inter-$^1$CT excitations develops *below* the intra-$^1$CT excitations. Namely, the second nearest-neighbour inter-$^1$CT states with $r_{e-h}$ ~1 nm are virtually degenerate with their inter-$^3$CT counterparts and in close energy resonance with the intra-$^1$CT states, thus enabling the interconversion between the intra-CT and second nearest-neighbour inter-CT manifolds. We anticipate that more distant (third-, fourth-, … neighbours) inter-$^1$CT pairs will be thermally accessible in larger clusters, on a par with the experimentally measured $r_{e-h}$.

Most importantly, this analysis is consistent with the comparison between the calculated and measured HFI-mediated singlet-triplet conversion rates. We have used DFT calculations to explore the HFI couplings in the BF2 inter-CT states, retaining both the isotropic Fermi contact and the dipolar tensor terms. The nucleus-dependent hyperfine magnetic fields were weighted according to the atomic contributions to the attachment (for anions) and detachment (for cations) densities and arranged to depict a (full) inter-CT excitation, see SI. The calculated HFI couplings, in the range of a few mT (~0.1–0.2 µeV), are in line with expected values for organics and consistent with our experimental findings. We next follow Fay and Manolopoulos to derive rate equations for the HFI-mediated spin conversion from second-order perturbation theory[56,57]. In this limit, the spin conversion rate scales as the square of the HFI coupling and is sensitive to the relative magnitude of the dephasing rate versus energy splitting between the initial and final states (see SI for further information on the model). We use the distance-dependent inter-CT state exchange energy, $J$, from TDDFT calculations in combination with the computed hyperfine couplings to predict HFI-ISC rates as a function of dephasing timescales, $\gamma$, in Fig. 3e. The HFI-ISC rate rapidly increases with increasing $r_{e-h}$ because of the lower singlet-triplet exchange energy and, assuming a reasonable range of dephasing times (estimated experimentally to be 40 ns to 1 µs, see SI for details; represented by the dashed vertical white lines), it reaches an order of magnitude approaching the experimental result of $10^7$ s$^{-1}$ at electron-hole pair separations of ≥1 nm. We stress that this occurs in a regime at the edge of validity of the perturbative model (Supplementary Fig. 43), suggesting the possibility for a coherent process.

In contrast, similar calculations performed on a representative nearest-neighbour 4CzIPN pair (extracted from the single crystal structure[42], see SI for details) yield degenerate singlet and triplet inter-CT states >0.12 eV above the lowest-lying intramolecular excitations (Fig. 5b); a relative energetic ordering opposite to that predicted in BF2. This is expected based on the near spherical symmetry of the 4CzIPN molecules that isolates the central A group of one molecule from the D moieties of neighbouring molecules. This raises the effective intermolecular $r_{e-h}$ in the closest molecular dimer to 0.76 nm: a value significantly larger

than the corresponding $r_{e-h}$ of 0.43 nm in the intramolecular exciton. As a result, the intramolecular excitons of 4CzIPN are not expected to readily dissociate into inter-CT states, matching our experimental observations in this material.

## Discussion

From the combination of our experimental and computational results, we can rationalise the spin interconversion processes in BF2 occurring after photoexcitation (Fig. 5c). In the context of OLED operation involving electrical excitation, where inter-CT states will be initially generated in a spin-statistical manner when the free electrons and holes come within their Coulomb capture radius[1,58], the 25% loosely bound and long-lived dark inter-$^1$CT states created can act as a reservoir for the formation of the slightly higher-lying and emissive intra-$^1$CT exciton. The 75% of injected carriers that form inter-$^3$CT states branch between: (i) conversion into inter-$^1$CT via HFI-ISC; (ii) dissociation into free charge carriers that are then recycled in the recombination loop[59]; (iii) recombination into lower-lying intra-$^3$CT excitons. We find little experimental evidence for the latter pathway in BF2; despite the clear presence of molecular triplet excitons in the PLDMR of the neat and 10 wt% APDC-DTPA, TXO-TPA, and 4CzIPN films (Supplementary Figs. 21–25), we only see a small intra-$^3$CT response in neat BF2 (Supplementary Fig. 19), and this is absent in BF2 at 10 wt% in CBP (Supplementary Fig. 20). Our calculations suggest that intra-$^3$CT/inter-$^3$CT electronic couplings are as strong as the ones obtained in the spin-singlet manifold (Supplementary Tables 10 and 14), but the intra-$^3$CT/inter-$^3$CT energy gap is much larger (~0.29 eV). Therefore, we speculate that near degeneracy of the intra-$^1$CT and inter-$^1$CT facilitates their interconversion, while this is hindered in the triplet manifold due to the relative stabilisation of the intra-$^3$CT. As a result, rISC via the HFI or the scrambling of the inter-CT state spins through a few cycles of inter-CT state dissociation and recombination will be favoured. Furthermore, the presence of an intramolecular rISC pathway in BF2, albeit slow, means that recombination to the intra-$^3$CT is also not necessarily a terminal event.

In summary, we have shown that it is possible to reconcile the conflicting requirements of a very strong optical absorption and emission with efficient spin state interconversion in aggregated DF molecules. We achieve this by utilising high oscillator strength intramolecular excitations for interaction with photons, and exploiting loosely bound inter-CT states, with a vanishingly small $\Delta E_{ST}$, for rapid ISC processes mediated by the HFI. We find evidence that the formation of inter-CT states can occur at very low emitter doping concentrations (4 wt%), resulting in a significant increase in the film PLQE; this suggests that HFI-ISC can effectively contribute to the emission processes in the highly dilute active layers typically employed in DF OLED devices. Whilst we have focussed here on OLED applications, our findings are also of relevance to OSCs. For example, the generation of loosely bound charge transfer states, and even free charge carriers, in neat BF2 films without the inbuilt driving energy provided by offset molecular orbitals in D:A blends will facilitate the development of efficient OSCs that do not require separate electron D and A materials. In addition, our study of BF2 provides an insight into the mechanism of the ultra-low driving energy charge generation seen in nonfullerene acceptor (NFA) OSCs, where it has been proposed that hole transfer process from the NFA to the polymer occurs via an inter-CT type state similar to those we report in BF2[47,60,61].

## Methods

**Transient absorption spectroscopy**. TA samples were fabricated by spin-coating solutions onto quartz substrates using identical conditions to the previously-reported optimised OLED devices[10]. The samples were encapsulated in a nitrogen glovebox environment to ensure oxygen-free measurements.

TA was performed on a setup powered using a commercially available Ti:sapphire amplifier (Spectra Physics Solstice Ace). The amplifier operates at 1 kHz and generates 100 fs pulses centred at 800 nm with an output of 7 W. A non-colinear optical parametric amplifier (NOPA) was used to provide the tuneable ~100 fs pump pulses for the ultrafast (100 fs–1.8 ns) TA measurements, whilst the second harmonic (532 nm) of an electronically triggered, Q-switched Nd:YVO$_4$ laser (Innolas Picolo 25) provided the ~1 ns pump pulses for the nanosecond-microsecond (1 ns–700 μs) TA measurements. The probe was provided by a broadband visible (525–775 nm) and near infrared (830–1000 nm) NOPAs. The probe pulses are collected with a Si dual-line array detector (Hamamatsu S8381-1024Q), driven and read out by a custom-built board from Stresing Entwicklungsbüro. The probe pulse was split into two identical beams by a 50/50 beamsplitter; this allowed for the use of a second reference beam, which also passes through the sample but does not interact with the pump. The role of the reference was to correct for any shot-to-shot fluctuations in the probe that would otherwise greatly increase the structured noise in our experiments. Through this arrangement, very small signals with a $\frac{\triangle T}{T} = 1 \times 10^{-5}$ could be measured.

**Time-resolved (ns-μs) PL**. Time-resolved PL spectra were recorded using an electrically gated intensified CCD camera (Andor iStar DH740 CCI-010) connected to a calibrated grating spectrometer (Andor SR303i). Sample excitation with a 532 nm pump pulse was provided by a NOPA. Temporal evolution of the photoluminescence emission was obtained by stepping the ICCD gate delay with respect to the excitation pulse. The steady-state PL spectra were measured using the CW mode of the spectrometer.

**Photoluminescence-detected magnetic resonance spectroscopy**. Samples for PLDMR were prepared by thermal evaporation (APDC-DTPA, TXO-TPA and 4CzIPN) and spin-coating (BF2) onto thin glass microscope cover slides. The substrates were then cut up into ~3 mm thick strips, stacked into quartz EPR tubes and sealed in a nitrogen glovebox with a bi-component resin (Devcon 5-Minute Epoxy), such that all PLDMR measurements were performed without air exposure.

PLDMR was measured at the Swedish Interdisciplinary Magnetic Resonance Center (SIMARC) with a modified Bruker X-band ESR spectrometer (microwave frequency 9.2–9.84 GHz) with optical access for light excitation and detection. A super-high-Q microwave resonator was used with Q = 8000 ± 1000. The microwave power was amplified by AXEPR10 units with maximum power of about 3.3 W at 0 dBm, corresponding to an amplitude of the microwave magnetic field (**B$_1$**) of about 0.38 mT. **B$_1$** was deduced from the calibration sheet provided by Bruker. The samples were excited by a 405 nm solid-state diode laser with excitation power of 30 mW and a beam spot of 2 mm diameter. The PL was detected by a Si photodiode in conjunction with suitable long-wavelength pass filters. PLDMR was registered as a change of PL intensity upon the magnetic resonance condition. Lock-in detection was used by modulating **B$_1$** with a rectangular waveform at a frequency of 1–100 kHz. The measurements were performed at room temperature (293 K).

**OSC device fabrication**. Indium tin oxide- (ITO) patterned glass substrates were cleaned by acetone and isopropanol for 20 min each. The substrates were dried using compressed nitrogen, followed by oxygen plasma treatment for 10 min. The conventional architecture devices were made by spin-coating a layer of poly(3,4–ethylenedioxythiophene):poly(styrenesulfonate) (PEDOT:PSS, Clevios P VP Al 8043) at 3000 rpm for 40 s onto the ITO substrates in air, followed by annealing in air at 150 °C for 20 min. The active layer for BF2 devices was then spin coated on top of the PEDOT:PSS layer inside a nitrogen filled glovebox from a 20 mg/mL chloroform solution at 1500 rpm. For the 4CzIPN devices, the substrates were transferred to a nitrogen filled glovebox and pumped down under vacuum (<10$^{-7}$ torr) for the evaporation of a 70 nm active layer. Finally, a 10 nm thick Ca interlayer followed by a 100 nm thick Al electrode were deposited on top of the active layer for both sets of devices. An Angstrom Engineering Series EQ Thermal Evaporator was used for thermal evaporation. The electrode overlap area was 4.5 mm$^2$. The active area of the device was determined using an optical microscope.

**OSC device testing**. Photovoltaic characteristic measurements were carried out inside a N$_2$ filled glovebox. Solar-cell device properties were measured under illumination by a simulated 100 mW cm$^{-2}$ AM1.5 G light source using a 300 W Xe arc lamp with an AM 1.5 global filter. The irradiance was adjusted to 1 sun with a standard silicon photovoltaic cell calibrated by the National Renewable Energy Laboratory. No spectral mismatch correction was applied. A Keithley 2635 A source measurement unit was used to scan the voltage applied to the solar-cell between −2 to 1 V at a speed of 0.43 V/s with a dwell time of 46 ms. Scans were performed in both the forward and reverse directions, with no unusual behaviour observed. At least 16 individual solar-cell devices were tested for each system reported.

**Steady-state absorption**. Steady-state absorption spectra were measured using an HP 8453 spectrometer.

**Photoluminescence quantum efficiency measurements**. The PLQE was determined using method previously described by De Mello et al.[62]. Briefly, the laser and PL spectra were measured with a blank, direct sample excitation, and indirect sample excitation. By integrating the laser and PL spectra for each of these cases, we could determine the fraction of laser photons that were directly absorbed by the sample and emitted again. From this information, we calculated the PLQE. Samples were placed in an integrating sphere and photoexcited using a 520 nm continuous-wave laser. The laser and emission signals were measured and quantified using a calibrated Andor iDus DU420A BVF Si detector. Nitrogen samples were encapsulated in a nitrogen filled glovebox prior to measuring to ensure no exposure to the oxygen in ambient air. Air samples were exposed to ambient air for 24 h in the dark prior to measuring to ensure significant oxygen ingression into the films.

**Computational details**. BF2 dimers and tetramer and 4CzIPN dimers were optimised at the DFT level with the B3LYP functional[63], using empirical dispersions, and the 6–31 G(d) basis set for all the atomic species. Intermolecular distances $d_\perp$ were found to be 0.35 nm after the ground-state optimisation. Singlet and triplet excited-state energies of these aggregates were then computed at the TDDFT level, resorting to the Tamm-Dancoff approximation (TDA)[64]. In the dimers, these calculations were performed as a function of $d_\perp$ by keeping fixed one fragment and shifting the other along the long molecular axis. A *screened* range separated hybrid (SRSH)[65,66] approach was employed in such calculations by using the LC-ωhPBE functional[67] and the 6–311 + G(d,p) basis set, as implemented in Gaussian16[68].

Diabatization of electronic states in the aggregates was performed with the same functional and the 6–311 G(d,p) and 6–31 G(d) basis sets for the dimers and tetramer, respectively. Deconvolution of the lowest singlet and triplet TDDFT adiabatic states in terms of diabatic states was done using the Boys localisation scheme[69]. These calculations were performed with the Q-Chem package[70].

The HFI calculations have been performed on the BF2 anion and cation in the ORCA 4.2 package[71] employing the SRSH approach using the LC-ωhPBE functional and the EPR-III basis set. The HFI couplings for the triplet excited states were obtained by weighting the HFI of the anion and the cation by the attachment (i.e. electron) and the detachment (i.e. hole) densities computed on the BF2 dimer.

**Reporting Summary**. Further information on research design is available in the Nature Research Reporting Summary linked to this article.

## Data availability
The data that support the plots within this paper is available at the University of Cambridge Repository: https://doi.org/10.17863/CAM.76085.

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

## Acknowledgements

A.J.G. and R.H.F. acknowledge support from the Simons Foundation (grant no. 601946), the EPSRC (EP/M01083X/1 and EP/M005143/1), and the European Research Council (ERC) under the European Union's Horizon 2020 research and innovation programme (grant agreement No 670405). Y.P. acknowledges support from the Swedish Energy Agency (EM-48594-1) and The Swedish Research Council (VR-2017-05285). M.C., E.Z. and F.F. acknowledge financial support from Aix Marseille Université and CNRS. C.T. and D.C. thank the Basque Government (PIBA19-0004) and the Spanish Government MINECO/FEDER (PID2019-109555GB-I00) for financial support. G.L and D.B. thank the European Union's Horizon 2020 research and innovation programme under Marie Skłodowska Curie Grant agreement No. 722651 (SEPOMO). Computational resources in Mons and Namur (D.B. and Y.O.) were provided by the Consortium des Équipements de Calcul Intensif (CÉCI), funded by the Fonds de la Recherche Scientifiques de Belgique (F.R.S.-FNRS) under Grant No. 2.5020.11, as well as the Tier-1 supercomputer of the Fedération Wallonie-Bruxelles, infrastructure funded by the Walloon Region under Grant Agreement No. 1117545. Y.O. acknowledges funding from the FRS-FNRS under the grant F.4534.21 (MIS-IMAGINE). G.R. acknowledges a grant from the 'Fonds pour la formation à la Recherche dans l'Industrie et dans l'Agriculture' (F.R.I.A.) of the F.R.S.-F.N.R.S. P.J.C. and D.M.U. (EP/L01551X/1) acknowledge support from the EPSRC.

## Author contributions

A.J.G., F.F. and R.H.F. conceived the work. A.J.G. performed the TA, trPL, and PLQE measurements. Y.P. conducted the PLDMR measurements. C.T., G.L., G.R., D.C., F.C., Y.O. and D.B. performed the quantum-chemical calculations. M.C. and E.Z. synthesised and characterised BF2. D.M.L.U. and P.J.C. fabricated and tested the OSC devices. A.J.G., E.W.E. and L.-S.C. fabricated the thin film samples used for the measurements. W.M.C., B.H.D. and N.C.G. contributed to the discussion of the PLDMR data. N.C.G., F.F., D.B. and R.H.F. supervised their groups members involved with the project. A.J.G., D.B. and R.H.F. wrote the manuscript with input from all authors.

## Competing interests

The authors declare no competing interests.
