## [Peer Review File · Nature Communications]

Spontaneous exciton dissociation enables spin state interconversion in delayed fluorescence organic semiconductorsREVIEWER COMMENTS

Reviewer #1 (Remarks to the Author):

The authors analyze the intramolecular and intermolecular RISC processes by taking BF2 as an example, and confirm the intermolecular RISC process coexisting with intramolecular RISC through progressive experimental design. Some other TADF molecules are also studied to confirm the conclusion. The proposal of HFI-ISC is extremely innovative and provides new ideas for the design of high-efficiency DF molecules. I think it is an important work and deserves publication. The following comments should be addressed.

- 1) The PIA band at 950 nm is classified as a hole polaron. Is it possible that it can be classified as an electron polaron?
- 2) For neat films, both intra- and inter- RISC process may exist at the same time. Are there any indicators that can measure the proportions of these two processes?
- 3) In general, ISC process proceed via spin-orbit interactions would be promoted when the transition type of S1 and T1 state are different. So, for the ISC process, will its rate be affected by the same transition type of S1 and T1?
- 4) How about the impact of polarity of matrix on the mentioned process? such as using a polar host PPF?

Reviewer #2 (Remarks to the Author):

Triplet harvesting mechanisms are crucial to the operation of organic light emitting diodes and a matter of great interest in the photophysics and organic optoelectronics communities. In this work the authors report the characterization of a model red emitter BF2 showing delayed fluorescence (DF). The manuscript is well written, the work is original and expected to attract great interest from the photophysics and OLEDs communities. I, therefore, recommend publication after the following points are addressed.

In isolation the slow reverse intersystem crossing (RISC) occurs within 260 microseconds, due to a relatively large 0.2 eV energy gap between the singlet and triplet intramolecular CT states. However, in the aggregated films BF2 intermolecular charge transfer states are formed very rapidly with electron-hole distances larger than 1.5 nm. In the intermolecular CT states the ST energy gap is thus negligible and RISC is allowed to proceed via hyperfine interactions (HFI), occurring within 24 ns. Transfer back to the emissive singlet CT exciton, then enables efficient DF.

- 1) Significant shifts of the steady-state emission are observed with increasing concentration. Is this emission arising from only the intramolecular or intermolecular CT states or from both?
- 2) In the 10% BF2/CBP films the steady state emission peaks around 730 nm, and the DF perfectly matches the steady state fluorescence. Why is then the delayed regrowth SE observed in films of 10% BF2 dispersed in CBP not matching the entire emission spectra, and only showing an increased intensity signal around 700 nm?
- 3) The authors claim the formation of aggregates is required to promote the dissociation of intramolecular CT excitons, but as close packing is also required this might result in luminescence being quenched in neat films. Using, bulky groups or dispersing the emitter in a host might enhance PLQY, but it seems detrimental to the dissociation of intramolecular excitons into intermolecular excitons where HFI might be viable. Therefore, further confirmation for the relevance of the role HFI is needed. This must involve some sort of quantification of the DF contribution to the overall emission. In particular, the DF contribution to the overall emission needs to be evaluated as concentration changes. Certainly, if HFI has a significant role, the DF contribution should increase with concentration in a significant manner, and the DF contribution must be much larger than in the diluted 2% samples.
- 4) The temperature dependence of the DF should also be reported. Is the HFI mechanism expected

to be affected by temperature or not?

5) The dissociation of charge transfer states in films of organic semiconductors has been reported previously and is usually associated with the observation of DF decaying in a power law fashion. In this work the mechanism is validated using mostly PIA and PDLMR studies. PIA bands attributed to GSB peaking at 630 nm, SE at 740 nm and intramolecular 1CT at 1000 nm are clearly observed upon excitation in neat films, but not so easily seen in doped films. The SE and intra 1CT PIA bands decay in picoseconds but the GSB shows very little recover on the same timescale. Another PIA band at 950 nm is attributed to free polarons (holes), and grows on the same timescale that the intra 1CT and SE decays, suggesting intra 1CT dissociation is occurring to form polarons. Regeneration of the intra 1CT due to bimolecular recombination of free polarons and interconversion between inter 1CT and intra 1CT occurring in the microsecond timescale is observed when BF2 is dispersed in CBP. PDLMR studies show quenching of the PL when the population of the intermolecular 3CT0 is decreased by excitation of 3CT0 to 3CT+ and 3CT- intermolecular triplet states.

All the previous observations indicate the HFI mechanism is a plausible cause and plays a significant role in the observation of delayed fluorescence in BF2. However, no direct luminescence decays have been given. The SE band observed in PIA measurements are noisy and could be potentially affected by the overlapp with PIA bands. It would be beneficial if the direct record of luminescence decays could be given. Are PF and DF decaying both exponentially? And how is the PF and DF lifetime changing with concentration and temperature?

Reviewer #3 (Remarks to the Author):

The manuscript by Gillett et al. reports on a mechanism through which (known) emitters with a large oscillator strength can still exhibit efficient delayed fluorescence. It is demonstrated that in such emitters reverse intersystem crossing takes place via first creating loosely-bound intermolecular charge-transfer states, which is relevant for triplet harvesting in organic light-emitting diodes and might be relevant for organic solar cells which could also benefit from the generation of separated charge carriers.

The manuscript is well written and the findings are interesting. These results might accelerate the development of efficient thermally activated delayed fluorescence emitters. I have no further comments on the manuscript and recommend publication.

Reviewer #1 (Remarks to the Author):

The authors analyze the intramolecular and intermolecular RISC processes by taking BF2 as an example, and confirm the intermolecular RISC process coexisting with intramolecular RISC through progressive experimental design. Some other TADF molecules are also studied to confirm the conclusion. The proposal of HFI-ISC is extremely innovative and provides new ideas for the design of high-efficiency DF molecules. I think it is an important work and deserves publication. The following comments should be addressed.

We thank the referee for their kind words about our manuscript. We set out below how we have addressed the points they have raised.

1) The PIA band at 950 nm is classified as a hole polaron. Is it possible that it can be classified as an electron polaron?

To assign the hole polaron of BF2, we have used TA to study a 1:1 blend of BF2 and PCBM (Figure S5). To further support this assignment, we have fabricated organic solar cell devices using this blend which show a moderately efficient photovoltaic response (now included as Figure S6). Furthermore, closely related BF2 derivatives show similar behaviour when blended with PCBM (10.1021/acsnenergylett.7b00157). Therefore, we can confidently say that electron transfer from BF2 to PCBM occurs, leaving a hole behind on BF2. It is certainly possible there is a separate PIA for the electron located on a BF2 molecule in the inter-CT state. However, we do not observe this in our measured probe range between 500-1000 nm. Thus, we conclude that this PIA lies outside of our probe range or does not possess a large enough absorption cross section to be detected by TA. We have added additional text to the manuscript to clarify this point:

‘The new species with a PIA at 950 nm is the hole polaron on BF2, as seen in the TA of a BF2:PC₆₀BM 1:1 blend film (Fig. S5); here, the efficient photocurrent generation in a reference organic solar cell (OSC) device fabricated from this system confirms that electron transfer is induced from BF2 to PC₆₀BM (Fig. S6), leaving behind a hole on BF2.’

Figure S6: The light current density-voltage curve of the corresponding BF2:PC₆₀BM OSC device, taken under 100 mW cm⁻² AM1.5G illumination. The significant photocurrent generation confirms that electron transfer from BF2 to PC₆₀BM occurs efficiently, supporting the assignment in Fig. S5 of the PIA centred at 950 nm as the hole polaron of BF2.

2) For neat films, both intra- and inter- rISC process may exist at the same time. Are there any indicators that can measure the proportions of these two processes?

Deconvoluting the intra- and inter- rISC processes is not trivial due to the overlapping timescales of these processes; whilst techniques such as PLDMR can explicitly confirm or exclude the presence of intra-³CT and inter-³CT states *via* the characteristic linewidths of the triplet species (related to the EPR *D*-parameter), as well as ISC/rISC mediated by HFI through the demonstration of 'spin-locking' measurements, it cannot quantify them. However, in the case of BF2, it is possible to be more quantitative about the contributions due to its distinctive photophysical behaviour. As BF2 forms inter-CT states on picosecond timescales, it can be assumed that in more concentrated films this pathway dominates over slower processes, such as intramolecular ISC to the intra-³CT driven by SOC that typically takes place on nanosecond timescales. Thus, a large majority of excitations in BF2 that live long enough to undergo rISC will be intermolecular in nature due to the rapid charge transfer process. Furthermore, due to the very slow rISC in isolated BF2 molecules, it is expected that the much quicker intermolecular pathway will dominate. As a result, in BF2 at least, we can confidently say that the intermolecular rISC process is dominant in all cases except the most dilute films. We have added additional text to the manuscript to reflect this:

'Thus, due to the large ΔE_{ST} and slow rISC process of isolated BF2 molecules, we propose that HFI-ISC between inter-CT states is the dominant triplet to singlet interconversion process in OLED devices fabricated from BF2.'

In addition, this referee may also find our response to point #3 raised by referee 2 informative; here, we discuss new measurements we have performed on a detailed dilution series of BF2 in CBP. This enables us to provide an empirical quantification of the contribution from delayed fluorescence to the photoluminescence quantum efficiency (PLQE) as a function of dopant concentration through the oxygen quenching of the PLQE. As a result, we find that the fraction of the PLQE quenched by oxygen rises with increasing wt%, indicating a larger contribution to the DF from the inter-CT rISC process. We also find evidence that the more efficient inter-CT mechanism starts to contribute at 4 wt% BF2 in CBP due to the faster SE quenching observed in the TA and the greatly improved PLQE compared to the 2 wt% film.

3) In general, ISC process proceed via spin-orbit interactions would be promoted when the transition type of S₁ and T₁ state are different. So, for the ISC process, will its rate be affected by the same transition type of S₁ and T₁?

In our manuscript, we find that it is the hyperfine interaction, not spin-orbit coupling, that is the primary driver of ISC/rISC in aggregated BF2. This means that the rate of the ISC/rISC process is controlled by the dephasing time between the spins of the electrons in the inter-CT states, as induced by the magnetic coupling between the electron and nuclear spins (see equation S3). Therefore, as this hyperfine-driven ISC/rISC process does not involve spin-orbit interactions (which the referee correctly infers would be vanishingly small in the case of S₁ and T₁ with the same transition type), it will not be affected by the transition types of S₁ and T₁. Indeed, for the ΔE_{ST} to be

small enough for HFI-ISC to operate (\sim neV), the singlet and triplet states involves will have essentially the same inter-CT transition type.

4) How about the impact of polarity of matrix on the mentioned process? such as using a polar host PPF?

We agree that it is interesting to consider how the polarity of the host may affect the formation of inter-CT states in BF2 and thank the referee for both bringing this to our attention and for proposing PPF as a potential polar host to consider. As BF2 cannot be evaporated and must be solution processed, we have decided to work with the similar phosphine-oxide based host DPEPO instead. We note that DPEPO is significantly more polar than CBP, with a dielectric constant of 6.1 (10.1038/s42005-018-0101-9) compared to 3.5 (10.1016/j.orgel.2018.09.002). Ultrafast TA studies of BF2 in DPEPO reveal that the formation of inter-CT states still occurs efficiently, as shown by the faster rate of SE quenching upon increasing the wt% of BF2. Furthermore, using nanosecond TA, we find evidence for the creation of long-lived excitations with the potential to undergo ISC/rISC mediated by the HFI. This new data has been included in the SI in Figs. S16-S18, with the figures included below for the convenience of the referee. We have also added the following text to the manuscript, with additional discussion on the BF2:DPEPO films included in the SI on page S19:

Main text:

‘Similar behaviour is observed in the polar host DPEPO (Bis[2-(diphenylphosphino)phenyl]ether oxide), which has a higher dielectric constant of $\epsilon = 6.1^{17}$ compared to $\epsilon = 3.5$ for CBP¹⁸, implying that the surrounding dielectric environment does not have a significant effect on the dissociation of intra-¹CT excitons in BF2 (Figs. S16-S18).’

SI:

‘To determine the effect of the dielectric environment on the ability of BF2 intra-¹CT excitons to dissociate into inter-¹CT states, we have also investigated BF2 diluted in the polar host DPEPO; DPEPO has a significantly higher dielectric constant of $\epsilon = 6.1^{11}$, compared to $\epsilon = 3.5$ for CBP¹². To enable a direct comparison to our studies on BF2 diluted in CBP, we have also fabricated DPEPO films with 2, 10, and 40 wt% BF2. In the ultrafast TA of BF2 in DPEPO (Fig. S16), we observe similar spectral features to the CBP hosted films; all samples show a broad and positive sign feature between 600 – 750 nm, which we assign to the overlapping GSB and SE of BF2. When comparing the 2, 10, and 40 wt% in DPEPO films, we see that the SE decays more quickly as the BF2 concentration increases (Fig. S17), indicating the quenching of BF2 intra-¹CT excitons by intermolecular charge transfer to form inter-¹CT states. In the nanosecond TA of the BF2:DPEPO films (Fig. S18), we observe a long-lived BF2 GSB feature, which we attribute to the presence of inter-¹CT states. Thus, we conclude that the polarity of the host environment has only a limited influence on the ability of BF2 to form well-separated and long-lived inter-¹CT states that have the potential to undergo ISC/rISC mediated by the HFI.’

Figure S16: The ultrafast TA spectra and kinetics of BF2 at 2, 10 and 40 wt% in DPEPO. 600 nm excitation with fluences of 4.7, 1.7, and 2.0 $\mu\text{J cm}^{-2}$ were used, respectively.

Figure S17: The normalised ultrafast TA kinetics of the BF2 SE (720 – 750 nm) for the 2, 10, and 40 wt% films in DPEPO and the neat BF2 film. The lifetime of the SE falls rapidly as the doping concentration is increased, confirming that the increased contact between BF2 molecules results in faster intermolecular charge transfer.

Figure S18: The nanosecond TA spectra and kinetics of BF2 at 2, 10, and 40 wt% in DPEPO. 532 nm excitation with fluences of 377, 75.3, and 57.2 μ J cm^{-2} were used, respectively.

Reviewer #2 (Remarks to the Author):

Triplet harvesting mechanisms are crucial to the operation of organic light emitting diodes and a matter of great interest in the photophysics and organic optoelectronics communities. In this work the authors report the characterization of a model red emitter BF2 showing delayed fluorescence (DF). The manuscript is well written, the work is original and expected to attract great interest from the photophysics and OLEDs communities. I, therefore, recommend publication after the following points are addressed.

We thank the referee for their positive feedback on our manuscript. Below, we set out how we have addressed the points that they have raised.

In isolation the slow reverse intersystem crossing (RISC) occurs within 260 microseconds, due to a relatively large 0.2 eV energy gap between the singlet and triplet intramolecular CT states. However, in the aggregated films BF2 intermolecular charge transfer states are formed very rapidly with electron-hole distances larger than 1.5 nm. In the intermolecular CT states the ST energy gap is thus negligible and RISC is allowed to proceed via hyperfine interactions (HFI), occurring within 24 ns. Transfer back to the emissive singlet CT exciton, then enables efficient DF.

1) Significant shifts of the steady-state emission are observed with increasing concentration. Is this emission arising from only the intramolecular or intermolecular CT states or from both?

The referee raises a good point about the cause of the significant red shift of the BF2 PL with increasing concentration. Our quantum-chemical calculations suggest the presence of two 'bright' adiabatic singlet states in the BF2 dimer with significant oscillator strength that are likely responsible for the observed emission (Table S7). Whilst both bright states have a majority contribution from the diabatic intramolecular states, there is a significant component from the intermolecular states too. Therefore, it is certainly feasible that the red shift in emission could be related to an increased delocalisation of the bright state wavefunction over neighbouring molecules. In addition, as BF2 will have a significantly larger ground state dipole moment than the CBP host, it is also likely that as the BF2 concentration increases, the intra-¹CT excited states will be progressively more stabilised by the surrounding dipolar BF2 molecules (10.1016/S0009-2614(99)00580-1). This will also contribute to the red shift in the emission. Therefore, we conclude that a combination of these factors will be responsible for the experimentally observed red shift. We have included additional text on this point in the SI on page S11:

We attribute this emission broadening and red shift to two factors: 1. As BF2 has a significantly larger dipole moment than the CBP host, as the BF2 concentration increases, the intra-¹CT excitons will be progressively more stabilised by the surrounding BF2 molecules, leading to a red shift in the PL maxima⁸; 2. We find that in the BF2 dimer, there is significant delocalisation of the bright state wavefunction over the neighbouring molecule that could result in a broadening and red shift in the observed emission (Table S7). This can be attributed to the tendency of BF2 to form dimers in low wt% films, followed by larger aggregates at a higher wt%⁹.

2) In the 10% BF2/CBP films the steady state emission peaks around 730 nm, and the DF perfectly matches the steady state fluorescence. Why is then the delayed regrowth SE observed in films of 10% BF2 dispersed in CBP not matching the entire emission spectra, and only showing an increased intensity signal around 700 nm?

The referee makes an important observation about the position of the SE that regrows on microsecond timescales not matching the steady-state PL spectrum. To help explain this discrepancy, we have provided below figures showing the TA spectra of BF2 at 10 and 40 wt% in CBP, taken on picosecond timescales where the SE band is strongly visible (black), nanosecond timescales where only the GSB is present (red), and microsecond timescales where the SE begins to reform (blue). We notice that the SE observed in the TA of BF2 is generally blue shifted compared to the steady-state PL, peaking around 680 nm in the 10 wt% film and 720 nm in the 40 wt% film. The reason for this difference is not clear but could be related to the overlap with other spectral features. However, the spectral location of the SE observed in the TA on picosecond timescales agrees well with the new band that forms at the lower energy edge of the GSB on microsecond timescales, providing strong evidence that it is indeed the regrowth of the SE. We have included these new figures in the SI as Fig. S13, with additional discussion:

Figure S13: A comparison of the ultrafast and nanosecond TA spectra for the 10 and 40 wt% BF2 in CBP films. The SE feature at 1 – 2 ps overlaps closely with the spectral position of the additional positive sign feature that grows in at the low energy edge of the GSB on microsecond timescales. This confirms that the low energy band represents the regrowth of the SE, induced by the reformation of bright intra-¹CT excitons from inter-¹CT states.

3) The authors claim the formation of aggregates is required to promote the dissociation of intramolecular CT excitons, but as close packing is also required this might result in luminescence being quenched in neat films. Using bulky groups or dispersing the emitter in a host might enhance PLQY, but it seems detrimental to the dissociation of intramolecular excitons into intermolecular excitons where HFI might be viable. Therefore, further confirmation for the relevance of the role HFI is needed. This must involve some sort of quantification of the DF contribution to the overall emission. In particular, the DF contribution to the overall emission needs to be evaluated as concentration changes. Certainly, if HFI has a significant role, the DF contribution should increase with concentration in a significant manner, and the DF contribution must be much larger than in the diluted 2% samples.

We appreciate the referee's request that additional evidence in support the role of HFI-ISC in BF2 be included in our manuscript. To achieve this, we have extended our studies of the BF2 dilution series in CBP to include additional wt% loadings, especially in the more dilute (<10 wt%) regime; the films we have now investigated contain 2, 4, 6, 8, 10, 20, 40 and 100 (neat) wt% of BF2 in CBP.

To begin, we have measured the photoluminescence quantum efficiency (PLQE) of this extended dilution series for films encapsulated inside of a nitrogen environment and exposed to air for 24 hours in the dark (included as Fig. 4a in the revised manuscript; tabulated values in Table S3). Beginning with the nitrogen samples, we make the interesting observation that the PLQE increases significantly from 2 wt% to 6 wt%, before declining again. This is consistent with the previous report of BF2, where it was found that the 6 wt% in CBP samples demonstrated the best OLED device performance (10.1038/s41566-017-0087-y). In addition, we show that the quenching fraction of the PLQE in the samples exposed to oxygen rises as the BF2 doping wt% increases (Fig. S26 in the revised SI), though the propagation of the error from the PLQE of the low wt% samples (resulting from the weak absorbance and the measured 1% fluctuation in the laser power) blurs the trend for the samples under 10 wt%. Though it has previously been noted that incomplete oxygen ingress into the solid films makes it difficult to precisely quantify the DF contribution to the emission (10.1021/acs.jpca.0c10391), we can empirically demonstrate that the more concentrated films have a stronger DF component. We note that we have chosen to use PL quenching measurements rather than trPL to investigate the PF/DF ratio as the PF decay for BF2 is significantly faster than the time resolution of our ICCD, precluding an accurate determination of this ratio; see the response to point #5 for more detail.

Fig. 4a:

Fig. S26:

To better understand the photophysical behaviour underpinning the unusual PLQE trends, we have performed additional TA studies on the BF2 films in nitrogen (now included as Fig. 4b in the revised manuscript; corresponding TA spectra are presented in Fig. S27 of the revised SI). We find that even the dilute 4 wt% film exhibits a noticeably faster SE quenching rate than the 2 wt% film, the latter of which represents the behaviour of isolated BF2 molecules. As we have previously demonstrated that the SE quenching in BF2 is due to the formation of inter-CT states, we conclude that there is sufficient aggregation in films with 4 wt% (and higher) BF2 loadings to enable intermolecular charge transfer. This is likely due to the large ground state dipole moment of BF2 curcuminoid derivatives, which drives the formation of closely interacting dimers, even in dilute films (10.1021/jo400389h, 10.1038/s41566-017-0087-y).

Fig. 4b:

Figure S27: The ultrafast TA spectra and kinetics of BF2 at 2, 4, 6, 8, 10, 20, and 40 wt% in CBP and a neat BF2 film. 600 nm excitation with fluences of 2.9, 2.3, 1.5, 1.2, 1.1, 0.8, 0.6, and 1.5 $\mu\text{J cm}^{-2}$ were used, respectively. These TA measurements were conducted on the same films used for the ‘nitrogen’ PLQE measurements. The TA kinetics are plotted in Fig. 4b.

To explain this behaviour, we propose that in the 2 wt% films, the clear bi-exponential decay of the BF2 GSB, with a long 'delayed' lifetime of 260 μ s (Figure S11b), represents the intramolecular ISC/rISC mechanism of isolated BF2 molecules driven by spin-orbit interactions. Here, the relatively low PLQE is primarily ascribed to non-radiative losses occurring in the triplet manifold due to the slow and inefficient rISC process. When moving from 2 wt% to 6 wt%, the enhanced intra-¹CT quenching and increased PLQE suggests the activation of the intermolecular HFI-ISC mechanism, where the rapid interconversion between inter-¹CT and inter-³CT states reduces the non-radiative losses in the triplet manifold. Finally, further increasing the doping concentration results in a reduction of the PLQE again, likely due to the combined effects of longer-range charge separation over larger BF2 aggregates, which is associated with non-radiative recombination losses (10.1038/s41467-019-13736-8), as well as the enhanced non-radiative decay rates accompanying the strongly red shifted emission. Thus, through these additional measurements, we believe we have satisfied the referee's request for additional confirmation of the relevance of the HFI, as well provided a more detailed understanding of the ability of the HFI-ISC mechanism to operate in the highly dilute films typical of most DF OLED devices. We have added significant discussion to our manuscript on these additional results, the most prominent of which can be found on pages 9-11 of the main text:

'To better understand the interplay between the intra- and inter-molecular rISC mechanisms BF2, we have performed further studies on a BF2 dilution series in CBP. We first note that the photoluminescence quantum efficiency (PLQE), measured in a nitrogen environment, rises from 44.3% to 63.3% as the doping fraction is increased from 2 to 6 wt%, before decreasing again (Fig. 4a, tabulated PLQE values in Table S3). We also find that the PLQE is consistently lower in films exposed to air than in those measured in a nitrogen environment, indicating a reduction in the DF contribution through the quenching of spin-triplet excitations by oxygen. Though the complete quenching of DF by oxygen in thin films is not expected due to incomplete oxygen penetration into the matrix³⁵, we make the empirical observation that the PL fraction quenched upon oxygen exposure rises as the BF2 concentration increases (Fig. S26), implying a larger DF contribution to the observed PLQE in higher wt% films. The trend in PLQE with doping fraction is consistent with the reported electroluminescence external quantum efficiencies (EQE_{EL}) of BF2 OLEDs, where the best performance was found for BF2 at 6 wt% in CBP¹⁰. Such behaviour is unusual for DF emitters, as the PLQE and EQE_{EL} are generally expected to fall as the doping fraction increases due to 'concentration quenching' effects³⁶. Furthermore, the PL maxima of the BF2 films shows a

strong red shift with increasing concentration (Fig. S8), meaning an additional reduction in the PLQE (and EQE_{EL}) is expected at higher wt% because of the energy gap law in organic emitters³⁷. This effect is expected to be particularly severe when the peak emission wavelengths shifts from 700 nm towards 800 nm³⁸, as occurs at higher BF2 loadings.

When examining the TA of the dilution series (Fig. 4b, corresponding TA spectra in Fig. S27), we find a clear reduction in the SE lifetime with increasing concentration, which we have attributed to the formation of inter-¹CT states from the bright intra-¹CT exciton. Indeed, the noticeable decrease in the SE lifetime when moving from 2 wt% to 4 wt% films suggests that there can be sufficient intermolecular interactions between emitter molecules to allow for the formation of inter-¹CT states at very low doping fractions; we note that the formation of inter-¹CT states at 4 wt% is also associated with a large increase in the film PLQE. Thus, we propose that in the 2 wt% films, the clear bi-exponential decay of the BF2 GSB, with a long ‘delayed’ lifetime of 260 μs (Figure S11b), represents the intramolecular ISC/rISC mechanism of isolated BF2 molecules driven by spin-orbit interactions. Here, the relatively low PLQE is primarily ascribed to non-radiative losses occurring in the triplet manifold due to the slow and inefficient rISC process. When moving from 2 wt% to 6 wt%, the enhanced intra-¹CT quenching and increased PLQE suggests the activation of the intermolecular HFI-ISC mechanism, where the rapid interconversion between inter-¹CT and inter-³CT states reduces the non-radiative losses in the triplet manifold. Finally, further increasing the doping concentration results in a reduction of the PLQE again, likely due to the combined effects of longer-range charge separation over larger BF2 aggregates, which is associated with non-radiative recombination losses³⁹, as well as the enhanced non-radiative decay rates accompanying the strongly red shifted emission.’

4) The temperature dependence of the DF should also be reported. Is the HFI mechanism expected to be affected by temperature or not?

We thank the referee for bringing this point to our attention. As the HFI-ISC ISC/rISC process relies on the magnetic coupling between the electron and nuclear spins (see equation S3), it is expected to be temperature independent. However, as the intra-¹CT is slightly higher in energy than the inter-¹CT (predicted to be 62 meV from our quantum-chemical calculations on the BF₂ dimer, Table S9), we would expect a temperature dependence in the reformation of the intra-¹CT and therefore the DF process in BF₂. We have added additional text on this topic to the SI on page S26:

‘As the HFI-ISC ISC/rISC process is driven by the magnetic coupling between the electron and nuclear spins (Equation S3), it is expected to be temperature independent. However, depending on the energetic alignment of the intra-¹CT and inter-¹CT states in the system of interest, a temperature dependence would be expected for the intra-¹CT dissociation ($E_{intra} < E_{inter}$) or inter-¹CT recombination ($E_{intra} > E_{inter}$) processes. For example, as the intra-¹CT is slightly higher in energy than the inter-¹CT in BF₂ (predicted to be 62 meV from our quantum-chemical calculations on the BF₂ dimer, Table S10), we would expect a temperature dependence for the reformation of the intra-¹CT after the cycles of ISC/rISC have taken place. Thus, the DF process in aggregated BF₂ samples is expected to be temperature dependent. This is consistent with previous observations in a 6 wt% BF₂ in CBP film⁹, where we expect both the intramolecular rISC (driven by spin-orbit interactions) and intermolecular HFI-ISC processes to be occurring.’

Whilst temperature dependent studies are certainly interesting, we believe that they are likely to yield complex results that are difficult to interpret. This is due to several factors, including: the general temperature dependence of the charge separation/recombination in organic semiconductors (10.1002/adfm.202107157), including the temperature sensitivity of the reorganisation energy for charge recombination (10.1021/bi952882y); the temperature dependence of the charge hopping process (10.1103/PhysRevB.81.045202), which is central to obtaining an electron-hole separation with a small enough ΔE_{ST} (~neV) for HFI-ISC to operate; and the temperature dependence of the host dielectric constant (10.1038/ncomms13680), which could affect charge separation/recombination by changing how well the charge carriers are screened from each other. Furthermore, as discussed in more detail in response to point #5 below, we are not able to resolve the ‘intermediate’ trPL timescales (hundreds of ns to μ s) for the BF₂ samples with our ICCD detector due to the extremely strong PF contribution. The absence of this time region would make it difficult to draw any solid conclusions from the temperature dependence of the trPL behaviour. For these reasons, we have decided against recording the temperature dependence of

the DF. However, we note that the previous report of BF2 does include some temperature dependent trPL data (Fig. S11 of 10.1038/s41566-017-0087-y), which shows a clear temperature dependence to the DF for the 6 wt% film of BF2 in CBP. We have therefore included additional discussion of this observation in the SI, also included in the text above.

5) The dissociation of charge transfer states in films of organic semiconductors has been reported previously and is usually associated with the observation of DF decaying in a power law fashion. In this work the mechanism is validated using mostly PIA and PDLMR studies. PIA bands attributed to GSB peaking at 630 nm, SE at 740 nm and intramolecular 1CT at 1000 nm are clearly observed upon excitation in neat films, but not so easily seen in doped films. The SE and intra 1CT PIA bands decay in picoseconds but the GSB shows very little recover on the same timescale. Another PIA band at 950 nm is attributed to free polarons (holes), and grows on the same timescale that the intra 1CT and SE decays, suggesting intra 1CT dissociation is occurring to form polarons. Regeneration of the intra 1CT due to bimolecular recombination of free polarons and interconversion between inter 1CT and intra 1CT occurring in the microsecond timescale is observed when BF2 is dispersed in CBP. PDLMR studies show quenching of the PL when the population of the intermolecular 3CT0 is decreased by excitation of 3CT0 to 3CT+ and 3CT- intermolecular triplet states.

All the previous observations indicate the HFI mechanism is a plausible cause and plays a significant role in the observation of delayed fluorescence in BF2. However, no direct luminescence decays have been given. The SE band observed in PIA measurements are noisy and could be potentially affected by the overlap with PIA bands. It would be beneficial if the direct record of luminescence decays could be given. Are PF and DF decaying both exponentially? And how is the PF and DF lifetime changing with concentration and temperature?

At the request of the referee, we have now included the trPL decays of the BF2 films studied in this work with the corresponding trPL spectra in Fig. S15. As expected, all samples in CBP show a long-lived DF contribution, though this is not resolvable in the neat film, likely due to the low PLQE. However, we would like to offer a word of caution about the overinterpretation of this data as our ICCD setup is not well-suited to measuring the distinctive photophysical properties of BF2. Firstly, we find that the smallest reliable ICCD gate step/integration time on our setup is 5 ns; from previous experience on our setup, smaller step intervals can induce artefacts into the data. As most TADF emitters show a PF lifetime on the order of ~10-20 ns, this time resolution is generally sufficient for accurately capturing the PF. However, we find that our ICCD setup is not appropriate for BF2, as its extremely high oscillator strength (the intra-CT absorption coefficient is a factor of 15 higher than the archetypal emitter 4CzIPN, see Table S1) results in a very short radiative lifetime; the intrinsic SE lifetime for isolated BF2 is ~2.4 ns (now included as Fig. S14). As this is significantly faster than our 5 ns ICCD time resolution, we are unable to fully resolve the PF contribution. In addition, owing to the intense PF in BF2 due to the large radiative rate, the DF appears as a much lower intensity relative to the PF in the trPL measurements. As a result, we cannot resolve the 'intermediate' time region from hundreds of ns to μ s as the ~50 ns integration time used for this section of the PL decay profile is not sufficient to detect the emission; the PL only becomes measurable again for the final section of the PL decay where longer integration times of tens of μ s are used. As such, we find that trPL measurements are of limited use for exploring the properties of BF2. Therefore, have chosen to focus on the TA measurements in this manuscript, as they are better suited to probing the photophysics of BF2.

Figure S15: The transient PL spectra of BF2 at 2, 10, and 40 wt% in CBP and a neat BF2 film. 532 nm excitation with a fluence $27.8 \mu\text{J cm}^{-2}$ was used for all samples.

Figure S14: A combination of the ultrafast and nanosecond TA kinetics for the SE feature of the 2 wt% BF2 in CBP film. The SE decay can be fitted with a mono-exponential function with a time constant of 2.4 ns, providing a proxy for the radiative lifetime of the BF2 intra-¹CT exciton that is not possible to obtain accurately from the trPL measurements due to the limited experimental time resolution of ~ 5 ns. Such a fast radiative rate is consistent with the extremely strong optical absorption of the BF2 intra-¹CT exciton.

Reviewer #3 (Remarks to the Author):

The manuscript by Gillett et al. reports on a mechanism through which (known) emitters with a large oscillator strength can still exhibit efficient delayed fluorescence. It is demonstrated that in such emitters reverse intersystem crossing takes place via first creating loosely-bound intermolecular charge-transfer states, which is relevant for triplet harvesting in organic light-emitting diodes and might be relevant for organic solar cells which could also benefit from the generation of separated charge carriers.

The manuscript is well written and the findings are interesting. These results might accelerate the development of efficient thermally activated delayed fluorescence emitters. I have no further comments on the manuscript and recommend publication.

We thank the referee for taking the time to review our manuscript and are pleased to see that they recommend its publication.

REVIEWER COMMENTS

Reviewer #1 (Remarks to the Author):

Authors have well addressed the comments and the paper quality has been improved. I recommend the publication of paper at this stage.

Reviewer #2 (Remarks to the Author):

I am satisfied with the way the authors responded to the questions made by reviewers. They made a great effort to clarify the questions as much as possible. Therefore, I recommend the paper is accepted in the current form.